# Exploring the Role of Video Playback Visual Cues in Object Retrieval Tasks

**DOI:** 10.3390/s24103147

**Published:** 2024-05-15

**Authors:** Yechang Qin, Jianchun Su, Haozhao Qin, Yang Tian

**Affiliations:** Guangxi Key Laboratory of Multimedia Communications and Network Technology, School of Computer, Electronics and Information, Guangxi University, Nanning 530004, China; yechangq@st.gxu.edu.cn (Y.Q.); 2113391057@st.gxu.edu.cn (J.S.); haozhaoqin@st.gxu.edu.cn (H.Q.)

**Keywords:** object discovery, indoor location, egocentric visual navigation, first-person video, mixed reality simulation

## Abstract

Searching for objects is a common task in daily life and work. For augmented reality (AR) devices without spatial perception systems, the image of the object’s last appearance serves as a common search assistance. Compared to using only images as visual cues, videos capturing the process of object placement can provide procedural guidance, potentially enhancing users’ search efficiency. However, complete video playback capturing the entire object placement process as visual cues can be excessively lengthy, requiring users to invest significant viewing time. To explore whether segmented or accelerated video playback can still assist users in object retrieval tasks effectively, we conducted a user study. The results indicated that when video playback is covering the first appearance of the object’s destination to the object’s final appearance (referred to as the destination appearance, DA) and playing at normal speed, search time and cognitive load were significantly reduced. Subsequently, we designed a second user study to evaluate the performance of video playback compared to image cues in object retrieval tasks. The results showed that combining the DA playback starting point with images of the object’s last appearance further reduced search time and cognitive load.

## 1. Introduction

Searching for objects is a common task in daily life and work that can be time-consuming [1,2]. Existing augmented reality (AR) systems often employ visual cues to guide users in object retrieval tasks. High-end AR devices (e.g., Hololens 2 [3], Vision Pro [4], etc.) equipped with spatial perception systems can provide 3D visual cues, accurately indicating object positions for users. For AR devices without spatial perception systems (e.g., Xreal air [5], Vuzix Blade [6], etc.), the image cue of the object’s last appearance serves as a common search assistance. Compared to static image cues, dynamic videos not only offer richer contextual information but also provide procedural guidance, potentially surpassing static image cues in enhancing users’ search efficiency. In addition, when a first-person video records the process of object placement, users watching the video playback can naturally orient themselves to the object’s location, thereby reducing users’ workload.

Despite these advantages, complete video playback capturing the entire object placement process as a visual cue can be excessively lengthy, demanding significant viewing time from users. Although playing only a portion of video playback or accelerating video playback can reduce users’ viewing time, it may lead to the loss of essential context information. This raises a research question: “After segmenting and accelerating the video playback of the object placement process, can it still effectively assist users in locating objects?” Thus, our research aims to address this question and optimize the effectiveness of video playback visual cues. In this work, we focus on three key events that occurring sequentially during the object placement process, which we use as starting points for video playback, selectively showcasing a portion of the object placement process. These key events can be described as follows: (i) object grasped (OG): when the object is picked up; (ii) destination appearance (DA): when the object’s final destination first appears in the video; and (iii) object released (OR): when the object is placed down at the final destination. To explore the effects of video playback in object retrieval tasks, we conducted a user study, which includes two factors: the starting point for video playback (OG, DA, and OR) and the playback speed (1×, 1.5×, and 2×). The results indicated that when playing the playback at normal speed from the DA starting point until the object’s last appearance, search time and cognitive load were significantly reduced. Subsequently, we conducted a second user study to evaluate the performance of three visual cues in object retrieval tasks. The visual cue factor includes three levels: (i) last frame (LF): the image of the object’s last appearance; (ii) video: video playback starting from DA at normal speed; and (iii) video-LF: video playback starting from DA at normal speed and consistently displaying the last frame after completion of the playback. Our evaluation demonstrated that the video-LF method further reduced the search time and cognitive load.

## 2. Related Work

### 2.1. Search Assistance

In both virtual and real-world environments, many technologies utilize virtual information to guide users to specific points of interest [7]. The primary guidance methods utilize either 2D or 3D visual cues [8]. Chittaro et al. [9] explored the effectiveness of using 2D or 3D arrows in virtual spaces to navigate users. The experiment results showed that the 3D arrow is equally effective as the 2D arrow in assisting users. Baudisch et al. [10] introduced the Halo technique, which visually indicates the positions of off-screen objects by encircling off-screen objects with partially visible circular rings. The positions of off-screen objects are demonstrated by the radius of the virtual rings. However, when a larger number of targets are located in the same direction, halo arcs can overlap and blend together. To address the arc overlap issue, Gustafson et al. [11] proposed Wedge, which encodes object orientation and distance into wedge-shaped cues. Wedges prevent arc overlap by rotating or adjusting the aperture of Wedge until the arc overlap is resolved. Building upon technologies such as 3D Arrow, Halo, and Wedge, visual cues such as Halo3D [12], HaloAR/WedgeAR [13], and FlyingArrow [14] have emerged. Oshimi et al. [15] developed LocatAR, an innovative system that automatically registers and indicates objects’ positions in large spaces using 3D curves. The endpoints of these 3D curves correspond to the object grasping and placement locations. Grandi et al. [16] discovered that adding shadows to virtual objects can assist users in grasping them more quickly. Respectively, some other methods directly encode object positions into visual cues based on the radar metaphor [17], including 3D Radar [18] and EyeSee360 [19].

Both 2D and 3D visual cues require the spatial positioning of the object. Some methods achieve object localization by attaching additional sensors to the objects [20,21,22,23,24]. In the context of augmented reality research, these additional sensors often take the form of markers (e.g., ARTag [25]). Al-Kalbani et al. [26] utilized cameras and markers to determine the position of virtual objects, facilitating collaborative interaction among multiple users with virtual objects in augmented reality systems. Li et al. [27] introduced FMT, a system utilizing wearable cameras to document interactions between elderly individuals and marked daily objects. Despite the accurate item location these additional localization sensors provide, methods that rely on additional sensors often have limitations regarding object requirements (e.g., inability to attach RF/ID sensors to heated containers). Another method for obtaining object locations involves using computer vision algorithms or integrated depth sensors to develop an understanding of the real-world environment [28,29,30]. Amirjavid et al. [31] proposed a system that combines cameras with wireless sensors to achieve precise localization by recognizing items’ visual appearance and users’ daily activities. The first camera, positioned at a distance from the scene, captures general visual information from the entire area. The second camera focuses on a more specific area, while wireless sensors provide approximate angles between the objects. The system was evaluated in a virtual environment. This approach also has its limitations, as it requires significant computational resources to establish scene understanding, making it challenging to deploy on lightweight AR glasses.

For AR devices without spatial perception systems, the last appearance of an object is often employed as a visual cue to assist users in their search tasks. Tavakolizadeh [32] introduced Traceband, a lightweight and portable bracelet-mounted camera that keeps track of commonly used objects. Users can locate missing items via a web-based software portal. Funk et al. [33] proposed a system that employs maps, arrows, and images of the last appearance of objects at different distances to guide users in locating items. User feedback indicated that the majority of participants responded positively to the last appearance image cues. Reyes et al. [34] developed a retrieval system capable of identifying objects in first-person images, emphasizing the crucial role of visual perspectives. Yan et al. [2] presented CamFi, a system identifying and logging the individuals who frequently were observed near target objects. The system associates static images of objects with the individuals who might possess them, providing users with an alternative method for locating items. Similarly, Yagi et al. [35] proposed GO-Finder, a system that automatically identifies and logs items during the object placement process. When users need to find these items later, the system provides a timeline of last appearance images to aid in object retrieval tasks.

### 2.2. First-Person Vision Method

With the widespread adoption of smartphones and wearable devices, first-person videos have become the most commonly used data in visual lifelogs [36]. Using images and first-person videos to guide users to specific locations is a common method for indoor navigation. Chang et al. [37] provided empirical evidence suggesting that first-person videos not only outperform static image cues in navigation but also provide a more immersive navigational experience for participants with cognitive impairments. In a comparative study, Xu et al. [38] established the superior usability and effectiveness of first-person videos, highlighting their advantages over traditional navigation aids such as 2D maps and human guides. Additionally, Roy et al. [39] introduced Follow-My-Lead, a novel indoor video navigation technique. They discovered that segmenting videos through the video checkpoints and rapid video transitions between checkpoints not only makes navigation more manageable, but also balances the fidelity and effort for users. First-person videos are often used for long-term activity recording. Several methods have been proposed to recognize human–object interactions in first-person videos. Spriggs et al. [40] successfully analyzed cooking actions through the integration of inertial sensors and first-person videos, segmenting the video playback into distinct phases of the culinary process. Kitani et al. [41] introduced a stacked Dirichlet process mixture model, facilitating swift recognition, categorization, and indexing of actions within first-person videos. Wang et al. [42] proposed an efficient framework: instead of treating motion boundaries and motion magnitudes in first-person videos as noise, this framework leverages these cues as motion cues, effectively integrating target and motion features. Augmented reality systems can extract information from physical hand movements by recognizing gestures [43]. In our work, we employ the SlowFast network [44] to recognize the human–object interactions in video playback recordings of the process of object placement.

## 3. User Study 1

A line of research has explored the method of simulating AR experiments within VR environments [45,46,47,48], which is known as mixed reality simulation. Lee et al.’s work indicates that significant performance differences between mixed reality simulation and augmented reality only emerge in the most challenging and reasoning-intensive search tasks [49]. To control variables accurately and leverage limited space, we rendered a virtual office scene as an experimental environment.

**Participants**. We randomly recruited 24 participants (5 females and 19 males, aged between 20 and 27) from a local university via email system. Among these participants, seven had prior experience with VR. All participants were right-handed.

**Apparatus**. We used the Valve Index VR headset and controllers as our experimental hardware, and the experimental software was developed using the Unity engine. For videos that recorded the entire process of object placement, we employed the SlowFast networks, Fast RCNN networks, and the model pre-trained on the Meccano dataset [50] to pinpoint the key events when the object was grasped, released, and last disappeared. Additionally, we inverted the video from the last appearance of the object and employed OpenCV’s CSRT tracker to determine the moment that the final destination of the object entered the video. To control variables accurately and leverage limited space, we rendered a virtual office scene within a real-world space of 6 meters by 6 meters. The size of the virtual scene provided in virtual reality was also 6 meters by 6 meters. Participants entered and started their object retrieval task from the center of the virtual office scene. In this virtual scene, there are a total of 40 possible item destinations, including 16 file cabinets, 16 office desk drawers, and 8 office desk tabletops. Interactive virtual objects are modeled based on real-world objects. When participants attempt to grasp or release a virtual object, the Steam VR interaction system recognizes the user’s hand movements via controllers and triggers the Throwable component in the Unity engine, thereby synchronizing user’s actions within the virtual environment. The experimental environment is depicted in Figure 1.

**Task**. Before each trial begins, the target virtual object will appear at a specific location within the virtual environment. The procedure of the experiment for each condition is described as follows: (i) the participant first practiced searching for a practice object to get familiar with the starting point and speed; (ii) after completing the practice trial, participants returned to the initial position and were shown an example image of the target virtual object to memorize its appearance; (iii) participants pressed the trigger on the right controller, and the video playback of the target object placement process appeared until the video playback completed; (iv) participants were tasked with freely walking in the virtual environment based on the information provided by the video playback and using their right hand to grasp the target object as quickly as possible; participants could replay the video at any time by pressing the controller trigger; (v) once they grabbed the target object, participants return to the initial position for the next trial; (vi) the participant repeated steps (iii) to (v) until the whole task was completed.

**Experimental design**. In this study, we examined the effects of playback starting point and playback speed on object retrieval tasks. The playback starting point had three levels (OG, DA, and OR), the playback speed had three levels (1×, 1.5×, and 2×). Murphy et al. [51] explored the relationship between video playback speed and students’ exam performance. The results of their first experiment showed that 2× speed was the fastest playback speed acceptable to students. In the second experiment, they found that watching the video playback at 2× speed immediately before the exam significantly enhanced students’ exam performance. Mo et al. [52] observed that high-level students show a better learning effect when the video is play at 1.5× playback speed. Despite the increased cognitive load associated with higher speeds, students still achieved better learning performance. Based on these findings, the playback speed factors had three levels: 1×, 1.5×, and 2×. We use three key events as playback starting points, ensuring that video playback starting from the occurrence of a key event and continues until the object’s last appearance. These key events can be described as follows: (i) object grasped (OG): when the object is picked up; (ii) destination appearance (DA): when the object’s final destination first appears in the video; and (iii) object released (OR): when the object is placed down at the final destination. We designed three video playback starting points based on these key events, as shown in Figure 2. After the end of video playback, the footage of video would disappear to prevent participants from gathering information from a static image rather than the dynamic content of the playing video.

Before the the main sessions, participants were instructed to grab and place six objects at random locations. The purpose of this practice block was to establish an in-depth understanding of the environment among the participants. Additionally, the participants were briefed on the contextual meanings of different playback starting points under the guidance of the experimenters. Each participant completed nine sessions of search trials, covering all possible combinations of the two factors. The orders of the two factors were counterbalanced across participants using balanced Latin square designs. Each session started with a practice trial for the participants, subsequently followed by three formal trials. After completing three sessions, participants took a three-minute rest. The total experiment took approximately 35 minutes, excluding rest time. After the experiment, we conducted a semi-structured interview to gather insights into the participants’ object retrieval experiences throughout the experiment. This user study included a total of 24 participants × 3 playback starting points × 3 playback speeds × 3 search trials = 648 trials. The parametric dependent variables were (i) search time, which was the average duration it takes for participants to grab the target object after pressing the controller trigger for the first time under a specific condition; and (ii) replay count, which tracks the number of times a user chooses to replay video playback after the first viewing in a session. After completing all the sessions, participants were asked to fill out a workload assessment questionnaire based on the NASA-TLX scale [53], which is a scale used to assess task workload. It includes multiple dimensions such as mental demands, physical demands, time pressure, task complexity, self-assessment, and task success level. After completing the workload assessment questionnaire, participants were asked to fill out the Likert 7-point scale. The specific statements included in the survey were as follows (ranging from 1: Strongly Disagree to 7: Strongly Agree): (i) I believe that I could gather sufficient information when the video was played using this method (Sufficiency); (ii) I found the video using this method to be intuitive and comprehensible (Intuition); (iii) I felt fatigued when the video was played using this method (Fatigue); and (iv) I would prefer the video to be played using this method (Preference).

**Result**. Table 1, Table 2 and Table 3 presents the results. After checking the normality and homogeneity of variance of the search time and replay count, we found that the search and replay count neither exhibits homogeneity of variance nor follows a normal distribution. Given these conditions, we used the non-parametric Aligned Rank Transform (ART) test and post hoc Wilcoxon Signed Rank tests to analyze the search time and replay count. For the subjective rating, we applied the non-parametric Friedman test with post hoc Wilcoxon Signed Rank tests.

**Search Time**. There was no significant interaction between the effects of playback starting points and playback speeds factors ($F_{4,92}=1.834$, p=0.129). A significant difference was found between the playback starting point’s levels ($F_{2,46}=27.716$, p<0.001). No significant difference was found between the playback speed’s level ($F_{2,46}=2.196$, p=0.123). At playback speeds of 1× and 1.5×, the search times for the DA method were significantly shorter compared to the OG and OR methods. Additionally, at a 1.5× playback speed, the OR (M = 25.6 s, SD = 9.2 s) method resulted in significantly longer search times than DA (M = 17.2 s, SD = 4.4 s) and OG (M = 21.4 s, SD = 6.3 s).

**Replay Count**. There was a significant interaction between the effects of playback starting points and playback speeds factors ($F_{4,92}=9.545$, p<0.001). A significant difference was found between the playback starting point’s levels ($F_{2,46}=76.124$, p<0.001). A significant difference was also found between the playback speed’s level ($F_{2,46}=36.337$, p<0.001). Across all playback speeds, the replay count under the OR method was significantly higher compared to the other methods. At 1× speed, the replay count under DA (M = 0.29, SD = 0.3) was significantly smaller than those under OR (M = 2.78, SD = 2.1, *p* < 0.001). The replay count under OG (M = 0.5, SD = 0.5) was significantly smaller than those under OR (*p* < 0.001). At 1.5× speed, the replay count under OR (M = 4.11, SD = 1.8) was significantly higher than those under OG (M = 0.68, SD = 0.5, *p* < 0.001) and DA (M = 0.93, SD = 0.6, *p* < 0.001). At 2× speed, the replay count under OR (M = 5.00, SD = 3.2) was significantly higher than those under OG (M = 0.93, SD = 0.9, *p* < 0.001) and DA (M = 1.76, SD = 1.4, *p* < 0.001).

**Workload**. We found significant differences across the two factors ($\chi^{2}\left(2\right)\;=\;18.084$, p<0.001; $\chi^{2}\left(2\right)\;=\;34.200$, p<0.001). Through pairwise comparisons, significant differences in workload were found between the OG, DA, and OR methods. Similarly, significant differences were found between playback speeds of 1×, 1.5×, and 2×. The workload under the DA method (M = 6.7, SD = 2.6) was significantly smaller than those under the OR method (M = 11.7, SD = 3.6, Z = −3.95, p<0.001). The workload under the OG method (M = 7.2, SD = 2.6) was significantly smaller than those under OR (Z=−3.62, p<0.001). The workload under 1× speed (M = 6.0, SD = 2.6) was significantly smaller than those under the 2× speed (M = 11.7, SD = 3.6, Z = −4.20, p<0.001). The workload at 1.5× speed (M = 6.2, SD = 2.1) was significantly smaller than those under the 2× speed (Z = −4.11, p<0.001).

**Sufficiency**. We found significant differences across the two factors ($\chi^{2}\left(2\right)\;=\;16.841$, p<0.001, $\chi^{2}\left(2\right)\;=\;31.921$, p<0.001). The ratings of Sufficiency under OG (M = 5.5, SD = 0.6, Z = −2.17, p=0.04) was significantly higher than those under DA (M = 4.9, SD = 1.1). The ratings for OR (M = 3.2, SD = 1.4) were significantly lower compared to both the OG (Z = −3.53, p<0.001) and DA (Z = −2.97, p<0.01). Ratings of Sufficiency were significantly higher at 1× speed (M = 5.6, SD = 1.4) compared to both 1.5× (M = 5.2, SD = 1.0, Z = −2.17, p<0.01) and 2× speeds (M = 3.6, SD = 1.4, Z = −3.66, p<0.001). The Sufficiency score under 1.5× speed was also significantly higher than those under 2× speed (Z = −4.08, p<0.001).

**Intuition**. We found significant differences across the two factors ($\chi^{2}\left(2\right)\;=\;13.156$, p<0.001, $\chi^{2}\left(2\right)\;=\;27.639$, p<0.001). Participants’ ratings of Intuition did not show a significant difference between DA (M = 5.1, SD = 1.4) and OG (M = 5.1, SD = 1.1). The OR method (M = 3.2, SD = 1.7) received ratings that were significantly lower than both OG (Z = −3.27, p<0.01) and DA (Z = −3.06, p<0.01). The Intuitive scores under 1× speed (M = 5.2, SD = 1.1) were significantly higher than those at 2× speed (M = 3.2, SD = 0.8, Z = −3.89, p<0.001). The ratings under 1.5× speed (M = 0.9, SD = 1.1) was also significantly higher than those under 2× speed (M = 3.3, SD = 1.1, Z=−4.09, p<0.001).

**Fatigue**. We found significant differences across the two factors ($\chi^{2}\left(2\right)\;=\;10.659$, p<0.001; $\chi^{2}\left(2\right)\;=\;16.900$, p<0.001). The DA method (M = 3.3, SD = 1.1) received ratings that were significantly lower compared to both OG (M = 4.6, SD = 1.4, Z = −2.48, p=0.013) and OR (M = 4.4, SD = 1.6, Z = −2.60, p<0.01). The fatigue scores at 1× speed (M = 3.4, SD = 1.4) were significantly lower compared to 2× speed (M = 4.8, SD = 1.3, Z = −3.11, p<0.01), and scores at 1.5× speed (M = 3.3, SD = 1.0) were also significantly smaller than those at 2× speed (Z = −3.47, p<0.001).

**Preference**. We found significant differences across two factors ($\chi^{2}\left(2\right)\;=\;19.419$, p<0.001; $\chi^{2}\left(2\right)\;=\;18.886$, p<0.001). The ratings for DA (M = 5.3, SD = 1.1) were significantly higher than those for OG (M = 4.3, SD = 1.3, Z = −2.26, p<0.001) and OR (M = 3.3, SD = 1.1, Z = −3.29, p<0.001). Preference scores at 1× speed (M = 4.5, SD = 1.1) were significantly higher compared to 2× speed (M = 3.3, SD = 1.4, Z = −2.49, p=0.013), and scores at 1.5× speed (M = 5.0, SD = 1.3) were also significantly higher compared to those at 2× speed (Z = −3.79, p<0.001).

**Qualitative Feedback**. When questioned about the reasons for replaying the video, all participants indicated that it was to identify hard-to-distinguish objects of reference. When asked about which aspects of video playback helped them the most in their object retrieval tasks, most participants (19/24) reported that the objects of reference in the video were the most helpful feature. However, two participants emphasized the importance of being familiar with the virtual office scene. For the OG starting point, some participants (9/24) appreciated the comprehensive information provided from the start of the video playback. As one participant put it, “Playing the video from the first starting point provides me with enough information, I don’t need to think”. However, several participants (8/24) felt that the OR method provided overwhelming information. A participant mentioned, “Excessive video playback duration make me impatient and forget a part of video content”. Another participant suggested a preference for a more concise video playback, mentioning, “A full playback isn’t necessary; capturing key features like a pen holder or a painting on the wall in the video would provide sufficient cues”. For the DA starting point, some participants (11/24) highlighted the advantage of immediate exposure to the object’s final destination. One participant noted, “Starting from the second starting point quickly gives me a rough idea of the object’s location, allowing me to plan my route even before the video ends”. In contrast, participants encountered certain challenges associated with the OR starting point. A participant mentioned, “Starting the video from the third starting point makes it difficult to clearly see the reference objects beside the item, complicating the search”. Another participant felt that this method left them with almost no path information, requiring more effort to interpret the environmental context. Most participants (13/24) reported that the video playback speed had minimal impact on their retrieval experience. The rest of participants, however, found the 2× speed to be excessively fast, leading to a loss of video information. A participant summarized, “While slow video playback makes me lose patience, overly fast playback causes details missing”.

**Summary**. Compared to other video starting points, the DA starting point required less time across the three playback speeds. Additionally, at normal speed (1×) and 1.5× speed, DA had fewer replay counts. Participants also gave more favorable ratings to the DA starting point. While the speed of playback did not significantly impact the search time, there was a significant tendency among participants to replay the video multiple times as the speed increased. Furthermore, user subjective ratings were significantly lower at double speed (2×). Despite the superior Sufficiency ratings at normal speed compared to 1.5× speed, no significant differences were observed between these two playback speeds.

## 4. User Study 2

In this user study, we seek to explore whether combining the normal speed video playback starting from DA with images of objects’ last appearance can enhance the user’s object retrieval experience and improve search efficiency.

**Participants**. We randomly recruited 24 participants (4 females and 20 males, aged between 20 and 27) from a local university via email system. Among these participants, seven had prior experience with VR. All participants were right-handed.

**Apparatus**. The hardware setup and the virtual experimental environment were the same as those in Section 3.

**Task**. Before each trial began, the target virtual object appeared at a specific location within the virtual environment. The procedure of the experiment for each condition is described as follows: (i) the participant first practiced searching for a practice object to get familiar with the visual cue; (ii) after completing the practice trial, participants returned to the initial position and were shown an example image of the target virtual object to memorize its appearance; (iii) participants pressed the trigger on the right controller, and the visual cue appeared; (iv) participants were tasked with freely walking in the virtual environment based on the information provided by the visual cue and using their right hand to grasp the target object as quickly as possible; participants could replay the video at any time by pressing the controller trigger; (v) once they grabbed the target object, participants return to the initial position for the next trial; (vi) the participant repeated steps (iii) to (v) until the whole task was completed. The task is depicted in Figure 3.

**Experimental design**. In this study, we evaluate the performance of three types of visual cues in object retrieval tasks: (i) last frame (LF): static image of object last appearance; (ii) video: the normal speed video playback starting from DA starting point; and (iii) video-LF: video playback starting from DA at normal speed and consistently displaying the last frame after completion of the playback. Numerous studies have confirmed the effectiveness of images of objects’ last appearance in object retrieval tasks [33,35]. Inspired by these studies on extracting static images of objects’ last appearance to serve as visual cues, we utilized the static image of the object’s last appearance in the video playback as a visual cue, which was last frame of video playback (LF). For the video, drawing from the conclusions of Section 3, we chose to present visual cues through video playback, employing the DA starting and playing at normal speed (1×). For the video-LF visual cue, the static image of the object’s last appearance, which is the last frame of the video, remained visible until the user chose to replay the video.

Before the main sessions, participants were instructed to grab and place six objects at random locations. The purpose of this practice block was to establish an in-depth understanding of the environment among the participants. Each participant completed three sessions of search trials, covering all three conditions. The order of the conditions was counterbalanced across the participants using a balanced Latin square design. Each session began with a practice trial, followed by six formal trials. Participants took a two-minute rest after each session. Excluding rest time, the entire experiment took approximately 25 min to complete. After the experiment, we conducted a semi-structured interview to gather insights into the participants’ object retrieval experiences during the experiment. This user study included a total of 24 participants × 3 visual cues × 6 search trials = 432 trials. The parametric dependent variable was search time, which was the average duration it takes for participants to grab the target object after pressing the controller trigger for the first time under a specific condition. After the experiment, participants were asked to fill out a workload assessment questionnaire based on the NASA-TLX scale, alongside a 7-point Likert rating survey. The specific statements included in the survey were as follows (1: strongly disagree, 7: strongly agree): (i) I am confident that I could gather sufficient information when using this visual cue (Sufficiency); (ii) I think that the usability of this visual cue seem more acceptable (Usability); (iii) I think this visual cue make me felt fatiguing (Fatigue); and (iv) I prefer using this visual cue for locating objects (Preference).

**Result**. Figure 4 and Table 4 show the result. To examine the homogeneity of variance and the normal distribution of search time, we applied the Shapiro–Wilk and Levene’s tests. The results indicated that the variance in search time was homogeneous and conformed to a normal distribution. Consequently, we employed the one-way ANOVA with post hoc Bonferroni to analyze the search time. For other non-parametric dependent variables, we adopted the non-parametric Friedman test with post hoc Wilcoxon Signed Rank tests.

**Search Time**. We found a significant difference ($F_{2,46}$ = 121.0, p< 0.001) among the three conditions. Specifically, the search time for the video-LF method (M = 14.7, SD = 2.8) was significantly shorter than for both video (M = 17.3, SD = 2.4, *p* = 0.007) and LF (M = 26.9, SD = 4.8, p< 0.001). Moreover, the search time for the video method was also reduced compared to the LF method (p<0.001).

**Workload**. We found a significant difference ($\chi^{2}\left(2\right)$ = 30.471, p<0.001) among the three conditions. The workload score under the video-LF method (M = 4.6, SD = 1.4) was significantly lower than the video (M = 6.0, SD = 1.7, Z = −2.91, p<0.01) and LF (M = 10.6, SD = 3.2, Z = −3.62, p<0.001) methods. The workload under the LF method was significantly higher than the video method (Z = 3.62, p<0.001).

**Sufficiency**. We found a significant difference ($\chi^{2}\left(2\right)$ = 29.419, p<0.001) among the three conditions. The Sufficiency score under the video-LF method (M = 5.8, SD = 1.3) was significantly higher than those under the video (M = 5.3, SD = 1.3, Z=−3.03, p<0.01) and LF (M = 3.8, SD = 1.1, Z = −3.65, p<0.001) methods. The video method also significantly outperformed the LF method (Z = −3.54, p<0.001).

**Usability**. We found a significant difference ($\chi^{2}\left(2\right)$ = 25.480, p<0.001) among the three conditions. The Usability score under video-LF method (M = 5.9, SD = 1.2) was significantly higher than those under the video (M = 5.2, SD = 1.1, Z = −2.81, p<0.01) and LF (M = 3.8, SD = 1.1, Z = −3.33, p<0.01) methods. The simplicity score under the video method was significant higher than the LF method (Z = −3.27, p=0.01).

**Fatigue**. We found a significant difference ($\chi^{2}\left(2\right)$ = 13.579, p=0.011) among the three conditions. The Fatigue score under the video-LF method (M = 2.6, SD = 1.0) was significantly smaller than for the video (M = 3.5, SD = 1.1, Z = −2.34, p=0.019) and LF (M = 4.3, SD = 1.8, Z = −2.67, p<0.01) methods.

**Preference**. We found a significant difference ($\chi^{2}\left(2\right)$ = 32.523, p<0.001) among the three conditions. The Preference score under the video-LF method (M = 6.2, SD = 0.8) was significantly higher than those under the video (M = 4.8, SD = 0.6, Z = −3.61, p<0.001) and LF (M = 3.3, SD = 0.9, Z = −3.64, p<0.001) methods, and the video method also significantly outperformed the LF method (Z = −3.47, p<0.01).

**Qualitative Feedback**. Most participants (15/24) reported that when using the LF visual cue, they tended to explore and search all locations similar to those depicted in the image. One participant stated, “When the target object is in a drawer, I can only open each drawer similar to what’s shown in the picture to find it”. A majority of participants (20/24) recognized and appreciated the benefits introduced by the video-LF method. Half of participants (12/24) noted that attaching the object’s last appearance image at the end of video playback facilitated quickly recalling the video content. As reported by one participant, “Attaching the image at the end of the video saves the effort of replaying the video”. Another participant highlighted the efficiency of this approach, stating, “During the search, if I forget (the content of the video), the attached image cue allows me to quickly recall it, saving time and effort compared to re-watching the entire video”. Additionally, some participants (9/24) commented on the balanced integration of path information and object location details between the video and the object’s last appearance image. They perceived that the video provided dynamic path information and a general sense of the object’s location, whereas the object’s last appearance image offered precise details about the object’s location. As one participant pointed out, “While path information obtained from the video is unlikely to be forgotten, object location details can be quickly recalled using the attached image cue if they are forgotten”.

**Summary**. Leveraging a combination of segmented video playback and static images of the object’s last appearance proves to be a more effective strategy for object retrieval tasks. Compared to only relying on either images the of object’s last appearance or segmented video playback, the video-LF method significantly enhances the user experience.

## 5. Discussion and Future Work

**Balancing video fidelity and search efficiency**. When users search for objects using the image cue, they first select certain objects within the image cue as objects of reference, and then match these objects of reference with corresponding objects in the real-world environment. This correlation establishes a connection between the image cue and spatial context, allowing users to pinpoint objects’ location. Compared to image cues, video playback visual cues offer dynamic views of objects of reference, making it easier for participants to identify the objects of reference. In user study 1, we found that starting videos from the OG led to longer search times (24.7 s) at 1× speed. Additionally, some participants mentioned that watching the entire object placement process caused them to forget certain details. This highlights the importance of segmenting video playback for better search efficiency. However, starting videos directly from the OR led to higher replay counts (2.7) and longer search times (28.1 s). Similar results were observed with 1.5× and 2× speed playback. Although the speed factor did not significantly impact search time, the workload ratings (10.6) and fatigue ratings (4.8) were higher at 2× speed, and sufficiency ratings (3.6) and preference ratings (3.3) were lower. These results may be attributed to reduced information fidelity in segmented and accelerated video playback. Due to this reduced fidelity, participants tended to repeatedly rewatch videos to obtain and memorize information; such an unnatural operation diminished users’ search efficiency. If the video playback starts from DA, users can understand the essential contextual information more effectively by seeing the first appearance of object’s destination in the video; at the same time, the length of the video playback is reduced to a considerable extent. Thus, this approach achieves a better balance between video fidelity and search efficiency.

**Video cues provides rich information about paths to target destinations**. Rich path information in video playback serves as crucial directional guidance, pointing toward the final destination. In user study 1, we observed that the video playback starting from OG and DA, which provide path information, posed lower workloads on users compared to the OR starting point. Similarly, in user study 2, we found that the LF visual cue resulted in higher workload ratings (10.6) than video-based methods. Participants from user study 1 reported that they could navigate effortlessly when starting the playback from the OG, which suggests that video playback navigates well in object retrieval tasks. However, the effectiveness of this navigation in other scenarios requires further evaluation. If users are already very familiar part of an environment (e.g., users’ own homes), the information provided by the OR starting point may be sufficient enough to help them find objects effectively. Previous research has indicated that in multi-floor and multi-room environments, image cues alone are insufficient for effective navigation. Additional visual cues, such as maps and arrows, are required to help users locate objects [33]. Although video playback visual cues offer navigation and object location information simultaneously, the video playback start from DA may not be optimal due to the omission of certain path information, such as going to another floor or room. In the future, we aim to evaluate the combined effects of video visual cues and other visual cues specifically in the above-mentioned complex environments.

**Does low-fidelity video playback work for older adults?** Both user studies recruited participants from a local university within the age range of 20 to 27 years. Even when participants used extremely low-fidelity video playback, they were still able to quickly retrieve the target object by searching all possible locations based on limited information. However, both obtaining information from low-fidelity videos and searching all possible locations may be overly challenging for older adults. Some studies found that due to cognitive decay, videos for older adults should be broken into meaningful bits [54]. Therefore, when guiding older adults in retrieval tasks using video playback visual cues, a short video with multiple checkpoints should be employed instead of continuous video playback.

**The impact of forgetting on object retrieval efficiency**. Another unexplored factor related to users is memory. To isolate the user studies in evaluating the guidance effects of video playback, we excluded the forgetting factor in both user studies. In the real world, object retrieval tasks often involve forgotten objects. The longer the time since object placement, the fuzzier the user’s memory of the object’s location becomes. In this case, higher-fidelity video playback visual cues may provide more effective guidance by helping user recall the memory of object placement. Will video playback become more useful in searching objects as the user’s memory becomes fuzzier? Should we provide video playbacks with different levels of fidelity for different levels of forgetting? We will explore the impact of forgetting on object retrieval efficiency.

## 6. Conclusions

In this work, we evaluated the effects of video playback in object retrieval tasks. The video playback was segmented based on three key events. We conducted two user studies to evaluate the impact of various methods for altering video duration and employing different visual cues on object retrieval efficiency and user experience. In the first user study, we found that the DA playback starting point, which highlights the comprehensive information of the final destination, was both the most effective and preferred choice among participants. On the contrary, accelerating playback speed failed to enhance retrieval efficiency and increased user workload. In the second user study, we found that attaching a static image of the object’s last appearance at the end of the video playback was found to help participants complete the retrieval task more efficiently and naturally.

## Figures and Tables

**Figure 1 sensors-24-03147-f001:**
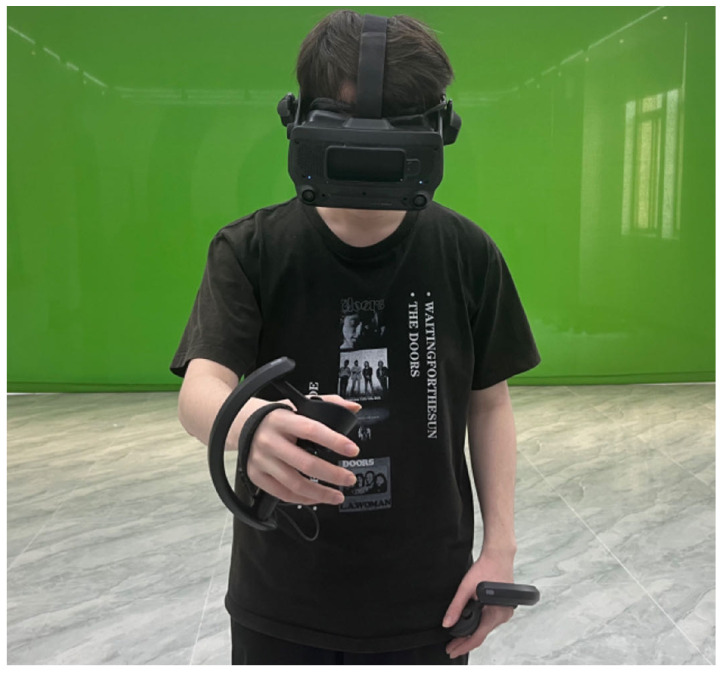
The participant is interacting with a virtual object in the experimental environment.

**Figure 2 sensors-24-03147-f002:**
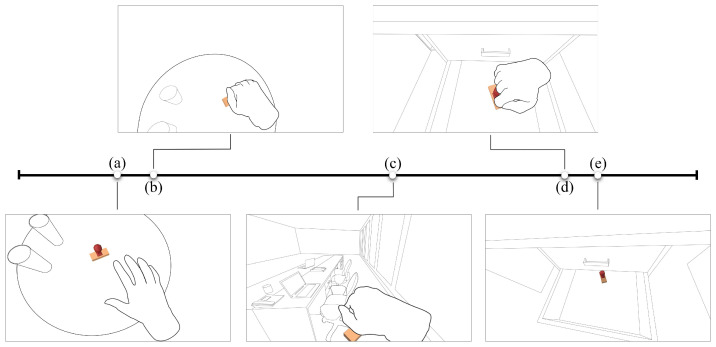
An example video timeline for user study 1. The white dots represent the video frames when creating different playback starting points. (**a**) The stamp is the target object for this example video. (**b**) OG playback starting point: This frame shows that object was grabbed by user. (**c**) DA playback starting point: This frame displays the container entering the camera’s field of view. (**d**) OR playback starting point: This frame shows that the object is being placed down at the destination. (**e**) The last appearance of the target object marks the end of the playback.

**Figure 3 sensors-24-03147-f003:**
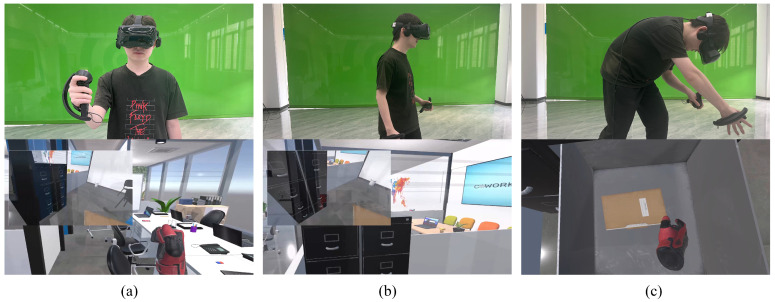
(**a**) The participant pressed the controller trigger and the video playback visual cue appeared. (**b**) Guided by the video playback visual cue, the participant approached the target object location. (**c**) The participant grabbed the target object.

**Figure 4 sensors-24-03147-f004:**
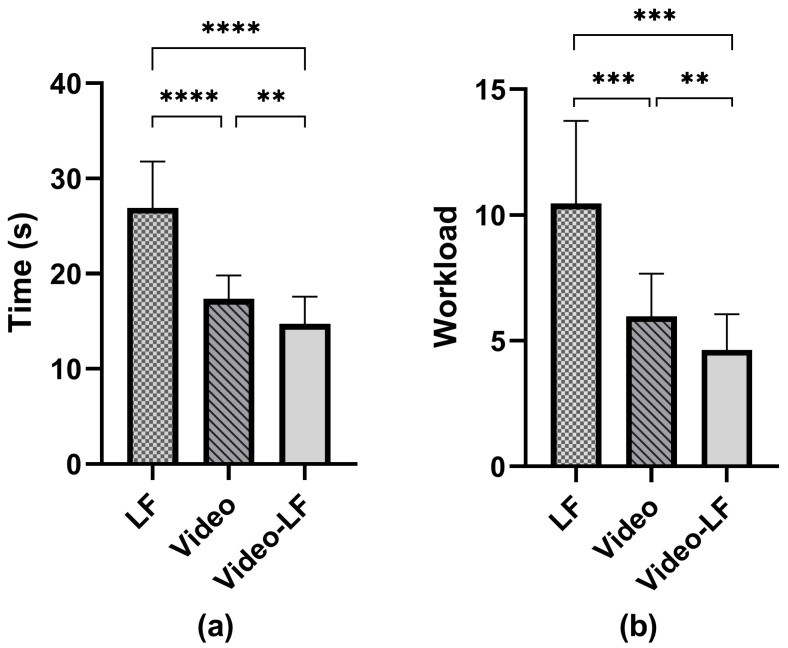
Search time and workload in user study 2. ** means *p* < 0.01; *** means *p* < 0.001; **** means *p* < 0.0001. (**a**) The mean search time. (**b**) The mean workload score based on the NASA-LTX form.

**Table 1 sensors-24-03147-t001:** The mean search time under nine conditions in user study 1: “# (#)” means the mean value and the corresponding standard deviation under a condition; “#–#” means that there was a significant difference between the numbered conditions.

Playback Speed	1 OG	2 DA	3 OR	Post-Hoc: Wilcoxon
1×	24.743 s (6.1 s)	17.782 s (4.5 s)	28.199 s (17.0 s)	1–2, 2–3
1.5×	21.424 s (6.3 s)	17.212 s (4.4 s)	25.622 s (9.2 s)	1–2, 2–3, 1–3
2×	22.860 s (8.3 s)	21.745 s (7.8 s)	27.987 s (7.6 s)	NA

**Table 2 sensors-24-03147-t002:** The mean replay count under nine conditions in user study 1.

Playback Speed	1 OG	2 DA	3 OR	Post-Hoc: Wilcoxon
1×	0.569 (0.5)	0.298 (0.3)	2.681 (1.9)	2–3, 1–3
1.5×	0.680 (0.6)	0.930 (0.6)	4.111 (1.8)	2–3, 1–3
2×	0.930 (0.9)	1.763 (1.3)	5.000 (3.2)	2–3, 1–3

**Table 3 sensors-24-03147-t003:** The workload and subjective ratings in user study 1.

**Subjective Rating**	**1 OG**	**2 DA**	**3 OR**	**Friedman Test**	**Post-Hoc: Wilcoxon**
Workload	7.2 (2.6)	6.7 (2.4)	11.7 (3.6)	$\chi^{2}\left(2\right)$ = 18.084, p<0.001	1–3, 2–3
Sufficiency	5.5 (1.4)	4.9 (1.1)	3.2 (1.4)	$\chi^{2}\left(2\right)\;=\;16.841$, p<0.001	1–2, 1–3, 2–3
Intuition	5.1 (1.4)	5.1 (1.1)	3.2 (1.7)	$\chi^{2}\left(2\right)\;=\;13.156$, p=0.001	1–3, 2–3
Fatigue	4.6 (1.4)	3.3 (1.1)	4.4 (1.6)	$\chi^{2}\left(2\right)\;=\;10.659$, p<0.001	1–2, 2–3
Preference	4.3 (1.3)	5.3 (1.1)	3.3 (1.5)	$\chi^{2}\left(2\right)\;=\;19.419$, p<0.001	1–2, 2–3, 1–3
**Subjective Rating**	**1 1×**	**2 1.5×**	**3 2×**	**Friedman Test**	**Post-Hoc: Wilcoxon**
Workload	6.0 (2.6)	6.2 (2.1)	10.6 (3.3)	$\chi^{2}\left(2\right)$ = 34.200, p<0.001	1–3, 2–3
Sufficiency	5.6 (1.4)	5.2 (1.0)	3.6 (1.4)	$\chi^{2}\left(2\right)$ = 31.921, p<0.001	1–2, 1–3, 2-3
Intuition	5.2 (1.1)	5.0 (1.1)	3.3 (1.1)	$\chi^{2}\left(2\right)\;=\;27.639$, p<0.001	1–3, 2–3
Fatigue	3.4 (1.3)	3.3 (1.0)	4.8 (1.3)	$\chi^{2}\left(2\right)$ = 16.900, p=0.001	1–3, 2–3
Preference	4.5 (1.1)	5.0 (1.3)	3.3 (1.4)	$\chi^{2}\left(2\right)$ = 18.886, p<0.001	1–3, 2–3

**Table 4 sensors-24-03147-t004:** The subjective ratings in user study 2.

Subjective Rating	1 LF	2 Video	3 Video-LF	Friedman Test	Post-Hoc: Wilcoxon
Sufficiency	3.3 (1.2)	5.5 (0.9)	6.4 (0.7)	$\chi^{2}\left(2\right)\;=\;29.419$, p<0.001	1–2, 1–3, 2–3
Usability	3.8 (1.7)	5.2 (1.1)	5.9 (1.2)	$\chi^{2}\left(2\right)\;=\;25.480$, p<0.001	1–2, 1–3, 2–3
Fatigue	4.3 (1.8)	3.5 (1.1)	2.6 (1.0)	$\chi^{2}\left(2\right)\;=\;13.579$, p=0.011	1–3, 2–3
Preference	3.3 (0.9)	4.8 (0.6)	6.2 (0.8)	$\chi^{2}\left(2\right)\;=\;32.523$, p<0.001	1–2, 1–3, 2–3

## Data Availability

Data are contained within the article.

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
