# Peer review of "Exploring the Role of Video Playback Visual Cues in Object Retrieval Tasks"

_sensors, 2024, doi:10.3390/s24103147_

Round 1
Reviewer 1 Report
Comments and Suggestions for Authors
In this paper, the authors explored the role of video playback visual cues in object retrieval tasks, especially for augmented reality (AR) devices without spatial perception systems. To explore whether segmented or accelerated video playback can still assist users in object retrieval tasks effectively, two user studies were carried out to compare the efficiency of segmented and accelerated playback against static image cues. The experimental results showed that when video playback is covering the first appearance of the object’s destination to the object’s final appearance (referred to as the Destination Appearance, DA) and playing at normal speed, search time and cognitive load were significantly reduced. Besides, combining the DA playback starting point with images of the object’s last appearance further reduces search time and cognitive load.
The exploration of video playback as a cue is novel, and it could significantly impact user interaction with AR applications. The paper is methodologically sound, and the two user studies are well-designed with a logical progression from testing individual video playback methods to comparing combined methods. The paper is presented clearly and concisely, and the experimental results can support the conclusions drawn. I recommend acceptance of this paper in present form.

Reviewer 2 Report
Comments and Suggestions for Authors
The paper investigates the effectiveness of utilizing video playback to assist users in object retrieval tasks, particularly in augmented reality (AR) devices lacking spatial perception systems. While images of the object's last appearance are commonly used for search assistance, the study explores whether videos capturing the process of object placement can offer procedural guidance, potentially enhancing users' search efficiency.
The study acknowledges that complete video playback covering the entire object placement process may be excessively lengthy, leading to significant viewing time investment. To address this concern, the researchers conducted a user study to explore whether segmented or accelerated video playback could still effectively assist users in object retrieval tasks. The results revealed that when video playback covers the object's destination appearance to its final appearance (referred to as Destination Appearance, DA) and plays at normal speed, both search time and cognitive load were significantly reduced.
Building on these findings, the researchers conducted a second user study to evaluate the performance of video playback compared to image cues in object retrieval tasks. The results demonstrated that combining the DA playback starting point with images of the object's last appearance further reduces search time and cognitive load.
Overall, the paper offers valuable insights into the potential benefits of using video playback as a tool for assisting users in object retrieval tasks, particularly in AR environments. However, several aspects need significant improvement before accepting the paper:
1. Clarity and Novelty: The paper could benefit from providing more detailed descriptions of the methodologies employed in the user studies, including the specific tasks assigned to participants and the metrics used to measure search time and cognitive load. Additionally, a deeper literature review of classical AR works should be conducted to provide a more comprehensive understanding of the research landscape. The following papers should be added and comparatively discussed: “Location-Based Augmented Reality Games Through Immersive Experiences. In: Schmorrow, D.D., Fidopiastis, C.M. (eds) Augmented Cognition. HCII 2021. Lecture Notes in Computer Science(), vol 12776. Springer, Cham”, “Collaborative manipulation of 3D virtual objects in augmented reality scenarios using mobile devices. 3DUI 2017: 264-265”, “Virtual Object Grasping in Augmented Reality: Drop Shadows for Improved Interaction. VS-GAMES 2019: 1-8”, and “A Review of Augmented Reality-Based Human-Computer Interaction Applications of Gesture-Based Interaction. In: Stephanidis, C. (eds) HCI International 2019 – Late Breaking Papers. HCII 2019. Lecture Notes in Computer Science(), vol 11786. Springer, Cham”. The deep literature review will make the paper more novel also.
2. Generalizability: While the results of the user studies are promising, the generalizability of the findings to other contexts or user populations may be limited. Providing insights into the characteristics of the study participants and the scenarios in which the experiments were conducted would help assess the broader applicability of the findings.
3. Discussion of Limitations: It would be beneficial for the paper to discuss any limitations or challenges encountered during the user studies, as well as potential avenues for future research. Addressing these aspects would provide a more comprehensive understanding of the study's implications and guide future research directions in this area.
In conclusion, the paper contributes valuable insights into the effectiveness of video playback as a tool for enhancing users' search efficiency in object retrieval tasks. However, further clarification of experimental methodologies, consideration of generalizability, and discussion of limitations are necessary to strengthen the paper's contribution to the field. With these enhancements, the paper has the potential to make a significant impact on research related to AR interface design and usability. Therefore, I recommend acceptance with suitable revisions to address the aforementioned points.
Comments on the Quality of English LanguageThe paper investigates the effectiveness of utilizing video playback to assist users in object retrieval tasks, particularly in augmented reality (AR) devices lacking spatial perception systems. While images of the object's last appearance are commonly used for search assistance, the study explores whether videos capturing the process of object placement can offer procedural guidance, potentially enhancing users' search efficiency.
The study acknowledges that complete video playback covering the entire object placement process may be excessively lengthy, leading to significant viewing time investment. To address this concern, the researchers conducted a user study to explore whether segmented or accelerated video playback could still effectively assist users in object retrieval tasks. The results revealed that when video playback covers the object's destination appearance to its final appearance (referred to as Destination Appearance, DA) and plays at normal speed, both search time and cognitive load were significantly reduced.
Building on these findings, the researchers conducted a second user study to evaluate the performance of video playback compared to image cues in object retrieval tasks. The results demonstrated that combining the DA playback starting point with images of the object's last appearance further reduces search time and cognitive load.
Overall, the paper offers valuable insights into the potential benefits of using video playback as a tool for assisting users in object retrieval tasks, particularly in AR environments. However, several aspects need significant improvement before accepting the paper:
1. Clarity and Novelty: The paper could benefit from providing more detailed descriptions of the methodologies employed in the user studies, including the specific tasks assigned to participants and the metrics used to measure search time and cognitive load. Additionally, a deeper literature review of classical AR works should be conducted to provide a more comprehensive understanding of the research landscape. The following papers should be added and comparatively discussed: “Location-Based Augmented Reality Games Through Immersive Experiences. In: Schmorrow, D.D., Fidopiastis, C.M. (eds) Augmented Cognition. HCII 2021. Lecture Notes in Computer Science(), vol 12776. Springer, Cham”, “Collaborative manipulation of 3D virtual objects in augmented reality scenarios using mobile devices. 3DUI 2017: 264-265”, “Virtual Object Grasping in Augmented Reality: Drop Shadows for Improved Interaction. VS-GAMES 2019: 1-8”, and “A Review of Augmented Reality-Based Human-Computer Interaction Applications of Gesture-Based Interaction. In: Stephanidis, C. (eds) HCI International 2019 – Late Breaking Papers. HCII 2019. Lecture Notes in Computer Science(), vol 11786. Springer, Cham”. The deep literature review will make the paper more novel also.
2. Generalizability: While the results of the user studies are promising, the generalizability of the findings to other contexts or user populations may be limited. Providing insights into the characteristics of the study participants and the scenarios in which the experiments were conducted would help assess the broader applicability of the findings.
3. Discussion of Limitations: It would be beneficial for the paper to discuss any limitations or challenges encountered during the user studies, as well as potential avenues for future research. Addressing these aspects would provide a more comprehensive understanding of the study's implications and guide future research directions in this area.
In conclusion, the paper contributes valuable insights into the effectiveness of video playback as a tool for enhancing users' search efficiency in object retrieval tasks. However, further clarification of experimental methodologies, consideration of generalizability, and discussion of limitations are necessary to strengthen the paper's contribution to the field. With these enhancements, the paper has the potential to make a significant impact on research related to AR interface design and usability. Therefore, I recommend acceptance with suitable revisions to address the aforementioned points.
Reviewer 3 Report
Comments and Suggestions for Authors
This paper concerns object retrieval through AR devices via video. The objective is to test whether segmented or accelerated video playback can effectively assist users in object retrieval tasks. Results indicated that when video playback covers the first appearance of the object's destination until the final appearance of the object and is played at normal speed, search time and cognitive load were significantly reduced. In a second study, the authors evaluated the performance of video playback versus picture cues in object retrieval tasks showing that combining the starting point of playback with pictures of the last appearance of the object further reduced search time and cognitive load.
---
The authors present a relevant topic for the scientific literature related to object search tasks in AR environments. However, some problems need to be addressed.
A. The motivation is general and not focused on the paper. The authors do not address the motivation concerning the problem identified and the solution they developed throughout the paper.
B. Authors did not formulate research questions within their paper, essential for conducting user studies.
C. The sample of participants is very limited (18 for user study 1 and 12 for user study 2). Explain why you chose this sample and what are the implications from the limitations point of view.
D. User study 2 does not present a representative picture of the task or users engaged in accomplishing it.
E. Sections 5 and 6 should be reformulated and have an organic narrative, rather than being divided into paragraphs detached from each other. Authors should discuss results more deeply, to make clear to the readers what was their aim and if they reached it through the experiments.
Other minor comments:
· In general, try to create a more organic text and avoid fragmentation of the various chapters, as the paper comes across as non-scientific if written this way.
· Acronyms: once stated in the paper, avoid repeating them in the extended form.
Round 2
Reviewer 3 Report
Comments and Suggestions for Authors
I commend the authors for the revised version of the manuscript which is greatly improved.
I wish you good luck with your future research!